


# The fate of O$^+$ ions observed in the plasma mantle and cusp: particle tracing modelling and Cluster observations

Audrey Schillings[1,2], Herbert Gunell[3,4], Hans Nilsson[1,2], Alexandre De Spiegeleer[3], Yusuke Ebihara[5], Lars G. Westerberg[6], Masatoshi Yamauchi[1], and Rikard Slapak[7]

[1]Swedish Institute of Space Physics, Kiruna, Sweden
[2]Division of Space Technology, Luleå University of Technology, Kiruna, Sweden
[3]Department of Physics, Umeå University, Umeå, Sweden
[4]Belgian Institute for Space Aeronomy, Brussels, Belgium
[7]EISCAT Scientific Association, Kiruna, Sweden
[6]Division of Fluid- and Experimental Mechanics, Luleå University of Technology, Luleå, Sweden
[5]Research Institute for Sustainable Humanosphere, Kyoto University, Japan, 611-0011, Gokasho, Uji, Kyoto

**Correspondence:** Audrey Schillings (audrey.schillings@irf.se)

**Abstract.** Ion escape is of particular interest for studying the evolution of the atmosphere on geological time scales. Previously, using Cluster-CODIF data, we investigated the oxygen ion outflow from the plasma mantle for different solar wind conditions and geomagnetic activity. We found significant correlations between solar wind parameters, geomagnetic activity (Kp index) and the O$^+$ outflow. From these studies, we suggested that O$^+$ ions observed in the plasma mantle and cusp have enough

energy and velocity to escape the magnetosphere and be lost into the solar wind or in the distant magnetotail. Thus, this study aims to investigate where do the ions observed in the plasma mantle end up. In order to answer this question, we numerically calculate the trajectories of O$^+$ ions using a tracing code to further test this assumption and determine the fate of the observed ions. Our code consists of a magnetic field model (Tsyganenko T96) and an ionospheric potential model (Weimer 2001) in which particles initiated in the plasma mantle and cusp regions are launched and traced forward in time. We analysed 136

observations of plasma mantle or cusp events in Cluster data between 2001 and 2007, and for each event 200 O$^+$ particles were launched with an initial parallel and perpendicular velocity corresponding to the bulk velocity observed by Cluster. From the observations, our results show that 93% of the events have an initial parallel velocity component twice the initial perpendicular velocity. After the tracing, we found that 96% of the particles are lost into the solar wind or in the distant tail. Out of these 96%, 20% escape into the dayside magnetosphere.

## 1  Introduction

Before the 1970's, it was believed that the solar wind was the primary source of magnetospheric plasma. However, this conception became obsolete a few years later with the studies of Shelley et al. (1976); Sharp et al. (1977) who observed ionospheric O$^+$ ions with high velocities in the high-latitude ionosphere. A few decades later, it is now well known that ion upflow from the ionosphere is a significant source for the magnetosphere (Hoffman, 1968; Chappell et al., 1987; Abe et al., 1993) and it is

accelerated through several processes to reach the high altitude cusp and plasma mantle. A part of this ion upflow is also known





as the polar wind in analogy with the solar wind (Axford, 1968). The polar wind is composed of $H^+$, $He^+$, $O^+$ and electrons with an energy of a few eV and commonly observed between $1000 \, \text{km}$ and roughly $50\,000 \, \text{km}$. Polar wind observations have been reviewed by Yau et al. (2007). At higher altitudes the terminology changes, and the term ionospheric outflow is used instead of polar wind, as it is complicated to distinguish solar wind $H^+$ from ionospheric $H^+$. Furthermore, the ion outflow is

then divided into two distinct populations, cold ions (up to a few tens eV) and hot ions (up to a few tens keV). The cold ions – detected with the spacecraft wake technique (Engwall et al., 2009) – are believed to be dominant for the magnetospheric plasma (André and Cully, 2012). These ions have been observed in the lobes (Haaland et al., 2009; Haaland et al., 2012; Liao et al., 2010) and have low enough parallel velocity so that convection dominates, and therefore will likely end up in the plasma sheet during strong solar wind conditions and southward interplanetary magnetic field (IMF) (Haaland et al., 2012). Under

northward IMF the convection is more stable and weaker Haaland et al. (2012), meaning this cold population will escape in the distant tail and be lost into the solar wind. The energetic ions are frequently heated transversely to the magnetic field due to wave-particles interactions in the cusp (Norqvist et al., 1998; Strangeway et al., 2005; Slapak et al., 2011; Waara et al., 2011; Nilsson et al., 2012) and parallel to the magnetic field due to centrifugal acceleration (Nilsson et al., 2008; Nilsson, 2011). Arvelius et al. (2005) showed that $O^+$ ions are accelerated from less than $0.1 \, \text{keV}$ to more than $1 \, \text{keV}$ between 8 and 12 Re.

The authors suggested that wave-particle interaction play the main role in the ion heating and subsequent acceleration. These energetic ions form the ion outflow at higher altitudes and several studies demonstrate the correlation between energetic ion outflow and solar and solar wind conditions (e.g., Peterson et al., 2001; Kistler et al., 2006; Li et al., 2012; Kistler and Mouikis, 2016; Schillings et al., 2019). It is the presence of a cusp and a polar cap that makes magnetised planets have atmospheric escape rates at least as high as planets without intrinsic magnetic fields (Gunell et al., 2018).

The main route of outflowing/escaping energetic ions is along open magnetic field lines, which include the polar cap, cusp and plasma mantle. The polar cap is defined as the footprint of the open magnetic field lines and the cusp as the entry of the solar wind into the magnetosphere. The plasma mantle is the region downstream of the cusp formed by reflected particles from the cusp, which are then convected toward the tail (Rosenbauer et al., 1975). Slapak et al. (2017) studied the $O^+$ outflow in the plasma mantle and dayside high-latitude magnetosheath for different geomagnetic conditions using the Kp index. They

found that the $O^+$ escape rate increases by 1.5 orders of magnitude during very disturbed magnetospheric conditions ($> \text{Kp}=6$) compared to quiet conditions ($\sim \text{Kp}=0\text{-}2$). Despite 5 years of data, Slapak et al. (2017) did not have enough statistics for extreme disturbances, and therefore Schillings et al. (2017, 2018) performed case studies of major geomagnetic storms ($> \text{Kp}=7+$) in order to complement the study of Slapak et al. (2017). The authors found a 2 orders of magnitude enhancement in the $O^+$ outflow for the major storms as compared to the average $O^+$ flux for the same year of each storm. They also suggested

that the $O^+$ ions have been heated enough when they reach the plasma mantle to eventually escape the magnetosphere. During major geomagnetic storms Slapak and Nilsson (2018) estimated a perpendicular velocity of the plasma mantle $O^+$ to $35 \, \text{km/s}$ and a parallel velocity of $115 \, \text{km/s}$, thus for their particular example the ions would reach the plasma sheet around -50 Re. As the near-Earth X-line is pushed towards Earth during disturbed conditions, these ions are expected to escape in to distant tail.

Models and simulations have been extensively employed to investigate polar wind and ionospheric outflow. Schunk and

Sojka (1989) simulated the polar wind behaviour using a combination of a low-altitude ionosphere-atmosphere and a high-



altitude hydrodynamic model in a simulated region from 120 km to 9000 km. They discovered the complexity of the polar wind density structures in different altitude ranges as well as for geomagnetic variations. Polar wind behaviour during one idealised geomagnetic storm has been investigated by Schunk and Sojka (1997), who updated their model to an altitude coverage of 90 km to 9000 km for latitudes higher than $50°$. They investigated the seasonal and solar cycle variations for four idealised

geomagnetic storms (winter and summer solstices and solar minimum and maximum). They found that $O^+$ upflow increases over the polar cap during the storms, while $O^+$ is the dominant ion species at all polar latitudes. These results are similar to the ones by Barakat and Schunk (2006) who studied the generalised behaviour of the polar wind, also during an idealised geomagnetic storm using a macroscopic PIC (particle-in-cell) model. Their results agreed with satellite observations. At an intermediate lower altitude of 4000 km, Horwitz et al. (1994) determined the bulk velocity and temperature profiles of $O^+$

and $H^+$ in the polar wind using a semi-kinetic outflow model. They found that centrifugal forces increase the outflowing $O^+$ flux with 2 orders of magnitude when the convection electric field is enhanced from 0 mV/m to 100 mV/m. A similar result has been shown by Abudayyeh et al. (2015), who used a Monte Carlo simulation based on the Tsyganenko T96 model and included the effects of the ambipolar electric field as well as gravitational and mirror forces. Additionally, Abudayyeh et al. (2015) observed higher bulk velocities and densities ($H^+$ and $O^+$) in the cusp than in the polar cap.

At an altitude range of 1.2 Re to 15.2 Re, Barghouthi et al. (2016) employed the same 1-D Monte Carlo model used by Abudayyeh et al. (2015) to investigate energetic $H^+$ and $O^+$ outflows along two trajectories (from the polar cap to the cusp) and compared them with Cluster data. Considering the centrifugal acceleration, the ambipolar electric field and the wave-particle interaction, they concluded that the latter was the most important mechanism especially at higher altitudes (cusp). Finally, a statistical model of the fate of energetic ions showed that these ions are highly dependent on the magnetic field

configuration. Therefore, for quiet magnetic field, more ions escape directly through the magnetopause, whereas for active magnetic field, the ions are convected towards the tail and reach the distant tail at 50 Re (Ebihara et al., 2006). Ebihara et al. (2006) also showed that under strong convection most of the ions in their model end up in the ring current.

Previous studies demonstrate that several models already exist to determine the behaviour of polar wind and/or ion outflow at different altitudes including the heating processes the ions are subject to. Ebihara et al. (2006) discussed the fates of the

ions launched at different magnetic local time (MLT) and at 1 Re. Furthermore, Krcelic et al. (2019) estimated the fate of ions using the Tsyganenko T96 model and observations of ion velocities observed by Cluster satellites. They suggested that 69% of the ions escape the magnetosphere with 50% in the distant tail. Despite all those interesting studies, the fate of ions observed in the plasma mantle has not yet been well defined. This study aims to clarify if $O^+$ ion outflows observed in the plasma mantle will escape the magnetosphere and be lost into the solar wind as suggested previously from observations (Slapak et al.,

2017; Slapak and Nilsson, 2018; Schillings et al., 2018). For a more accurate estimate of the fate of ions, the starting point should be high-altitude, so that much of the transverse heating and centrifugal acceleration are already included. In order to answer this question, we traced particles in a combination of the Tsyganenko T96 (Tsyganenko, 1995) and the Weimer 2001 (Weimer, 2001) models. About 25 000 $O^+$ particles were launched from the plasma mantle with initial parameters taken from Cluster observations. This model thus incorporates the effect of the mirror force on the launched ions, centrifugal acceleration





and $\boldsymbol{E} \times \boldsymbol{B}$ drift. It does not include any further wave particle interaction than what the ions had experienced prior to the observation point.

This paper is organised as follows: Section 2 describes the instrumentation and data set we used, followed by the method and a description of our code in Section 3. Section 4 and 5 present and discusses our results respectively. Finally, the final section, Section 6, summarises our study.

## 2 Instrument and data

### 2.1 Cluster and solar wind data

The Cluster mission (Escoubet et al., 2001) consists of four identical spacecraft flying in a tetrahedral formation with an apogee and perigee of approximately 19 Re and 4 Re respectively. On board the spacecraft, the Cluster Ion spectrometer (CIS) is composed of two instruments; the Hot Ion Analyser (HIA) and the COmposition and DIstribution Function analyser (CODIF)

(Rème et al., 2001). The latter provides 3-D distributions of ions with an energy resolution of $\Delta E/E \sim 0.16$, an energy per charge range between $25\,\mathrm{eV/q}$ and $40\,\mathrm{keV/q}$ and a $360°$ field of view. The resolution of the data is usually $4\,\mathrm{s}$, however, it can go up to $16\,\mathrm{s}$. Those features enable observations of $O^+$ in different magnetospheric plasma regions. Additionally, Cluster has a FluxGate Magnetometer (FGM) (Balogh et al., 2001) with a mode sample frequency of $22.4\,\mathrm{Hz}$. In our study, we use the magnetic field averaged over the spacecraft spin period ($4\,\mathrm{s}$).

The solar wind data were retrieved from the OMNIWeb database. This database consists of data from several satellites at diverse positions around Earth. In our simulations (see Section 3.2), we utilise the solar wind dynamic pressure and velocity, the IMF By and Bz components in high-resolution ($5\,\mathrm{min}$) as well as the magnetic Dst index ($1\,\mathrm{h}$).

### 2.2 Plasma mantle and cusp dataset

Our dataset consists of plasma mantle and cusp events observed by the Cluster spacecraft 4 between 2001 and 2007. In order

to only retrieve plasma mantle and cusp data, we apply several constraints on the observational data. Firstly, the CODIF $O^+$ counts are contaminated when strong proton fluxes from the magnetosheath are recorded at the same energy level as the $O^+$ ions (Nilsson et al., 2006). These false $O^+$ counts usually originate from the magnetosheath and lead to an underestimate of the $O^+$ velocity moment. The technique to remove these false counts is based on the $\boldsymbol{E} \times \boldsymbol{B}$ drift, because the drift is neither mass nor charge dependent. Consequently, using the kinetic energy equation, the cross talk signal is seen as an $O^+$ perpendicular

bulk velocity that is 1/4 of the corresponding perpendicular proton velocity, and typically the $O^+$ density is higher than $2\,\mathrm{cm}^{-3}$ (for more details, we refer the reader to Nilsson et al. (2006)). We avoid these contaminated data (and therefore magnetosheath data) using the method described by Nilsson et al. (2006). However, we slightly changed the threshold defined by Nilsson et al. (2006) because over the years the quality of the Cluster data devolved and so did the threshold. The new thresholds are given by $\frac{v_{tot(O^+)}}{v_{tot(H^+)}} < 0.2$ and $\frac{v_{tot(O^+)}}{v_{tot(H^+)}} > 0.5$ as well as $\frac{N_{O^+}}{N_{H^+}} > 0.25$.



To pick out only cusp and plasma mantle observations we implement different conditions for the high latitude regions. In these regions, the plasma beta $\beta$ (O$^+$ and H$^+$) is typically higher than 0.05, whereas it is lower than 0.05 in the polar cap (Liao et al., 2010, 2015; Haaland et al., 2017). We use a threshold value of $\beta > 0.1$ for high latitude regions. Additionally, the perpendicular temperature of the protons should be lower than 1750 eV in order to distinguish plasma sheet from plasma mantle data (Nilsson et al., 2006; Kistler et al., 2006; Slapak et al., 2017). As partly mentioned above, the O$^+$ and H$^+$ densities

are restricted to n(H$^+$) > 10$^{-3}$ cm$^{-3}$ and 10$^{-3}$ cm$^{-3}$ < n(O$^+$) < 2 cm$^{-3}$ to keep only reliable velocity estimates. In order to study the fate of ions, we take O$^+$ data with an outward flow (v$_\parallel$ > 0 or v$_\parallel$ < 0 in the southern and northern hemispheres respectively). Finally, we use a spatial coverage restriction to remove the inner magnetosphere, which is defined by -5 Re < X$_{\mathrm{GSM}}$ < 8 Re and $R_{\mathrm{GSM}} = \sqrt{Y_{\mathrm{GSM}}^2 + Z_{\mathrm{GSM}}^2} > 6$ Re (see also Slapak et al. (2017)). Major geomagnetic storms data are removed to exclude other magnetospheric regions than the cusp and plasma mantle (Schillings et al., 2017).

When all the above conditions are met, we define one event by 60 data points or more in a row. Between 2001 and 2007, our automatic routine detected 136 events that met the region criteria and the model restrictions (see Section 3.1).

## 3   Methodology

The section aims at briefly describing how our model works and its inputs and outputs.

### 3.1   Particle tracing simulations

We use a test particle simulation code (Gunell et al., 2019) to compute ion trajectories in the magnetic fields given by the Tsyganenko T96 model (Tsyganenko, 1995) and electric fields derived from the ionospheric potential given by the Weimer 2001 model (Weimer, 2001). The electric field is defined on a grid, and during the test particle trajectory calculation the electric field at the particle position is found by interpolation. Before the trajectory calculation starts, we define the electrostatic potential $V$ on a three-dimensional grid with a cell size of $1200\,\mathrm{km} \times 1200\,\mathrm{km} \times 1200\,\mathrm{km}$ in the region $-60 < X < 10$ Re,

$|Y| < 19$ Re and $|Z| < 19$ Re by tracing the magnetic field lines from each cell down to the ionosphere, where we retrieve the potential from the Weimer model. The electric field is then found from the relationship $\mathbf{E} = -\nabla V$. Figure 1 illustrates the magnetic field lines in light grey and the electric field grid in brown. This illustration represents the magnetosphere based on T96 and the grid for the interpolation of the electric field. Due to the limits of the Tsyganenko model, the electric field grid goes to -60 Re in the tail and 10 Re in the dayside (Tsyganenko, 1995). In the nort–south, $Z$, and dawn–dusk, $Y$, directions,

the limit of the grid are at $\pm 19\,\mathrm{Re}$. Note that the illustration is not to scale.

    To launch a particle, its initial ion velocity is calculated using the following equation

$$v_{tot} = v_\parallel \frac{\boldsymbol{B}}{B} + v_{\boldsymbol{E} \times \boldsymbol{B}} + v_\perp \frac{\boldsymbol{E}}{E}, \tag{1}$$

where $v_\parallel$ and $v_\perp$ are the parallel and perpendicular velocities of the O$^+$ ion respectively (see Section 3.2 for more details), $B$ the magnetic field, $E$ the electric field and $v_{\boldsymbol{E} \times \boldsymbol{B}}$ the $\boldsymbol{E} \times \boldsymbol{B}$ drift velocity in the model. Then, using the magnetic field line at

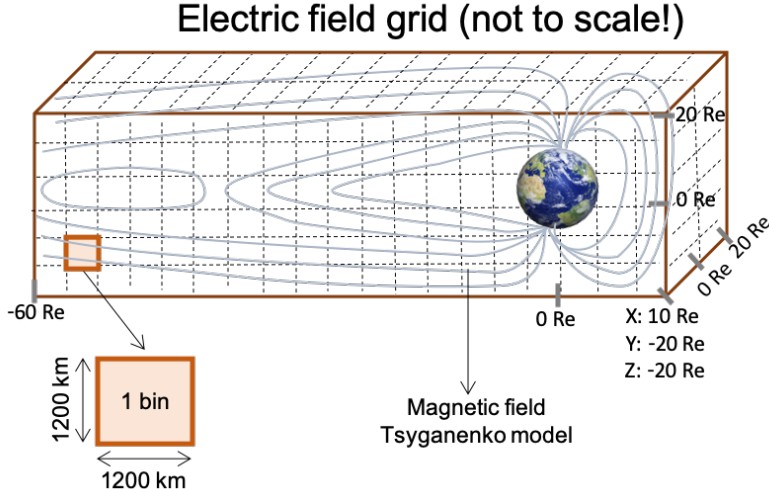

**Figure 1.** Schematic representation of the modelling of the Earth's environment. The Earth magnetosphere is represented in light grey and the brown grid displays the electric field grid. Note that the illustration is not to scale.

the position where the O$^+$ ion is launched, we interpolate the electric field in the corresponding electric cell. Finally, from the interpolated electric field, a new velocity is calculated with the Boris algorithm (e.g. Birdsall and Langdon, 1991).

The last step is repeated as far as the limitations of the code allows it. The tracing uses a time step based on the gyro-period, so that our time step is $dt = 2\pi/20\omega_c$ whereas the maximum number of iterations is limited to 10000. In 99,5% of the cases, the particle stops because they reach the limit of our model (electric field grid), whereas for the 0.5% remaining the maximum

number of the iterations have been completed.

The grid that defined the limit of our model is sufficient for our study because it includes the whole magnetosphere around Earth (magnetopause is defined around X = 10 Re in the dayside and about $|Y| = 13$ Re and $|Z| = 13$ Re). Further in the magnetotail, the magnetopause expands into a "cone shape" in the Y and Z direction and beyond 200 Re in the X direction. Our grid stops at $X = -60$ Re in the tail due to Tsyganenko model limits, moreover most of the particles reaching that distance will

most likely be lost (see Sections 4 and 5 for more details). Concerning constraints, the Weimer model imposes no constraints on solar wind parameters, while Tsyganenko T96 does. Therefore, when an observation meet the criteria described above it also has to match with Tsyganenko T96 constraints, which concerns the Dst index (-100 nT < Dst < 20 nT), the dynamic pressure (0.5 nPa < $P_{\mathrm{dyn}}$ < 10 nPa) and IMF $B_z$ and $B_y$ ($-10$ nT < $B_z, B_y$ < 10 nT).

### 3.2    Inputs and outputs of the model

The inputs of the models are (a) solar wind parameters as required by the Tsyganenko- and Weimer models, (b) the positions, $v_\parallel$ and $v_\perp$ based on Cluster observations. The solar wind parameters (see Section 2.1) are taken for each corresponding event. The 136 plasma mantle- and cusp events are automatically detected by a routine scanning Cluster data (see Section 2.2). During





these events, we calculate the bulk parallel and perpendicular velocities and retrieve the spacecraft positions. These parameters are then used to create the initial positions, $v_\parallel$ and $v_\perp$ of 200 $O^+$ ions (per event) that we trace forward in time. Note that the

perpendicular component of the velocity corresponds to the general variability of the data set. Instead, the thermal velocity could have been used, which would mostly lead to a larger range of perpendicular velocities, and in turn the mirror force would give even higher parallel velocities along most of the trajectories.

     Figure 2 shows the bulk parallel- and perpendicular velocities from Cluster data from a sample event in the northern hemisphere. This plasma mantle event occurred on 11$^{th}$ of June 2001 between 01:24 UT and 01:29 UT. The solid black line shows

the weighted mean defined by $\sum v_{\parallel,i} n_i / \sum n_i$ (where $i$ denote the observations, typically one 4 s measurement for CODIF), whereas the dashed red lines display the standard deviations. For this event, the mean $v_\parallel(O^+)$ is -109.01 $\pm$ 44.54 km/s and the $v_\perp(O^+)$ is 61.63 $\pm$ 36.71 km/s. A uniform standard distribution of random values in these intervals $v_\parallel$ = [-64.47;-153.55] km/s and $v_\perp$ = [24.92;98.34] km/s give the initial $v_\parallel$ and $v_\perp$ utilised as inputs for the forward traced particles. In a similar way, the initial positions of the 200 traced $O^+$ particles are randomly chosen in the interval x = [2.046;2.061], y = [-8.643;-8.558],

z = [8.885;8.886], which are the minimum and maximum positions of Cluster during the event (11.06.2001 - approximately 5 min).

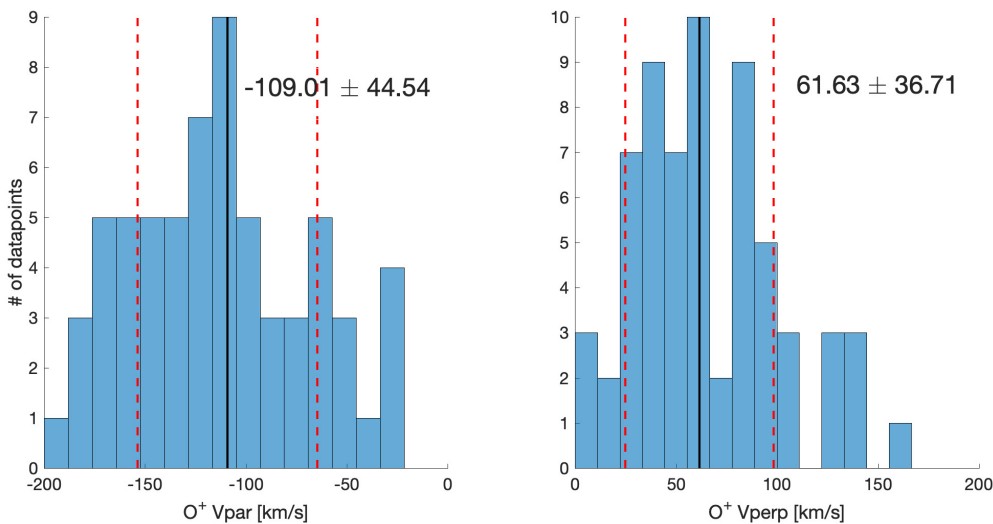

**Figure 2.** Cluster SC4 observations: Parallel and perpendicular components of the $O^+$ velocity on 11.06.2001 between 01:24 UT and 01:29 UT. The solid black line represents the mean and the dashed red lines show the standard deviations.

     The output of the model give us the final positions of $O^+$ in the magnetosphere as well as the travelling times of the particles in the magnetosphere.





## 4   Observations and Results

We analysed 136 event based on Cluster observations between 2001 and 2007. For each event, we launched 200 $O^+$ ions with various perpendicular and parallel velocities. Therefore, the statistics presented in this section are based on 27200 $O^+$ ions starting in the high altitude regions. Their ending positions are spread within the magnetosphere but a significant amount end up at the limit of our model. An example of 40 $O^+$ trajectories (out of the 200 computed) is shown in Fig. 3. This event occurred on 06.11.2001 during approximately 5 minutes (01:24 UT - 01:29 UT), the Dst index was -10 nT with a slightly

southward IMF (Bz = -0.486 nT) and negative By (-1.16 nT). The solar wind velocity was around 550 km/s and the dynamic pressure 1.74 nPa. The top panel in Fig.3 displays the trajectories in the XZ plane, while the middle and bottom panels display the YZ and XY planes respectively. The different colours represent various trajectories, the crosses show the initial positions (noted by starting positions), whereas the asterisks show the final positions. This event clearly shows that ions from similar positions but with different velocities (see Fig. 2 for velocity range) can have very distinct trajectories. Part of the $O^+$ ions

follow the magnetic field lines and stop at -60 Re in the distant tail (limit of the model), others are mirroring a few times before being lost on the flank (see bottom panel). Finally, a few ions are mirroring back and forth around Earth and end up in the cusp, the polar cap or simply in the plasma sheet. In this event (Fig. 3), 196 trajectories out of the 200 computed are considered to be long (more than 2000 iteration steps, see next paragraph for more details). In other events, we observed ions following magnetic field lines into the distant tail that eventually reach the plasma sheet around X = -50 Re and turn back toward Earth

(not shown). Those particles are return flow (earthward flow), and we discuss their fate in the Discussion (see Section 5).

Since the plasma mantle is close to the magnetopause, some events have very short trajectories (approximately 8 min). Indeed, the $O^+$ ions that are launched at high altitudes in the plasma mantle typically follow the magnetic field lines and reach the magnetopause almost immediately. Those ions escape into the magnetosheath and will most likely never turn back to the magnetosphere. Fig. 4a shows the length of the 27200 trajectories in our sample, the mean trajectory is about 1030 iteration

steps (or $10^{2.75}$ in the panel). We analysed the fate of the ions with short (lower than 200 steps, average time 8 min), middle (200 to 2000 steps, average time 25 min) and long (over 2000 steps, average time 130 min) trajectories and found that $O^+$ trajectories with less than 2000 steps (short and middle) escape mainly from the flank of the magnetosphere and represents 89% of our samples. Ions with longer trajectories represent 11% of the total sample. Within the ions with a long trajectory, 32% end up in the near-Earth plasma sheet (at geocentric distance lower than 10 Re). We defined the escaping limit by the geocentric

distance of the final positions $R = \sqrt{X_{\text{fin}}^2 + Y_{\text{fin}}^2 + Z_{\text{fin}}^2}$ that equals 10 Re. This is justified by the fact that if the tracing does not end due to the limits of iterations, such ions have left the simulation domain (except for 0.5% of the trajectories, see Section 3.1). Note that the minimum geocentric distance where the $O^+$ ions are launched is 7.64 Re (not shown). Only 4 % of the total trajectories have their final positions below this limit (10 Re), hence 96 % of the ions are escaping the magnetosphere. The geocentric distance $R$ of the 27200 final positions is represented in the middle panel (b) of Fig. 4. The $O^+$ average final

position is $R = 23.5$ Re.

Furthermore, we determined the minimum distance in the $X$ direction for each trajectory, see Fig. 4c. This parameter is important because some particles that interact with the plasma sheet in the distant tail might come back close to Earth.





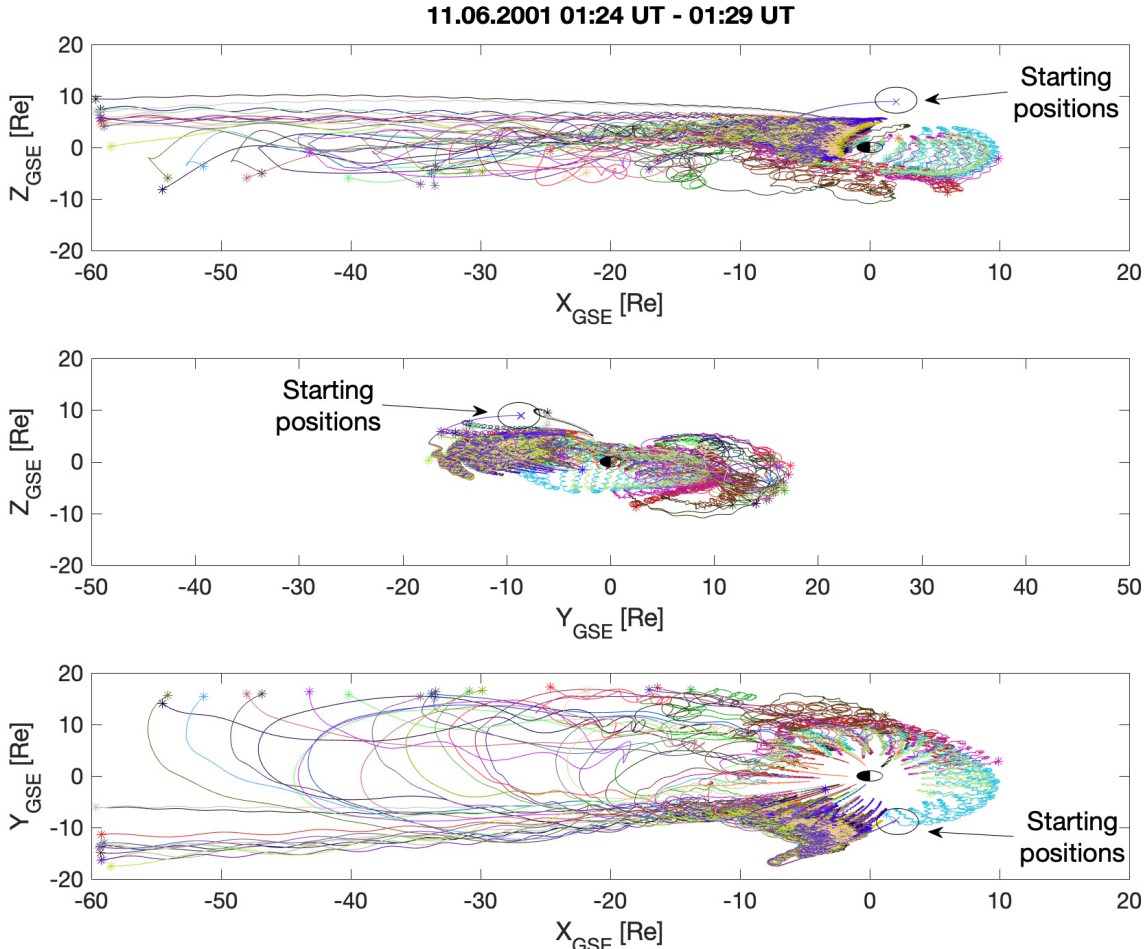

**Figure 3.** Example of 40 O$^+$ varied trajectories (different colours) from the plasma mantle on 11.06.2011 between 01:24 UT - 01:29 UT. The crosses denote the starting positions, whereas the asterisks denote the ending positions in the magnetosphere.

However, such cases are rare because for a total of 1751 trajectories having a X minimum distance beyond -50 Re only 79 trajectories finish their route close to Earth ($R < 10$ Re). The 1672 remaining are roughly equally spread between 10 Re and 66 Re. The average minimum X distance is around -10 Re, which corresponds to the plasma mantle region if |Z| > 5 Re (see also on Fig. 5).

Fig. 5 shows the start (left panel) and stop (right panel) positions of all trajectories in cylindrical coordinates ($R_{cyl} = \sqrt{Y^2 + Z^2}$). The colour bar represents the numbers of trajectories. In the left panel, we clearly see that particles are launched in the plasma mantle/cusp region, while on the right panel, the ending positions are spread at high $R_{cyl}$. O$^+$ ions from the plasma



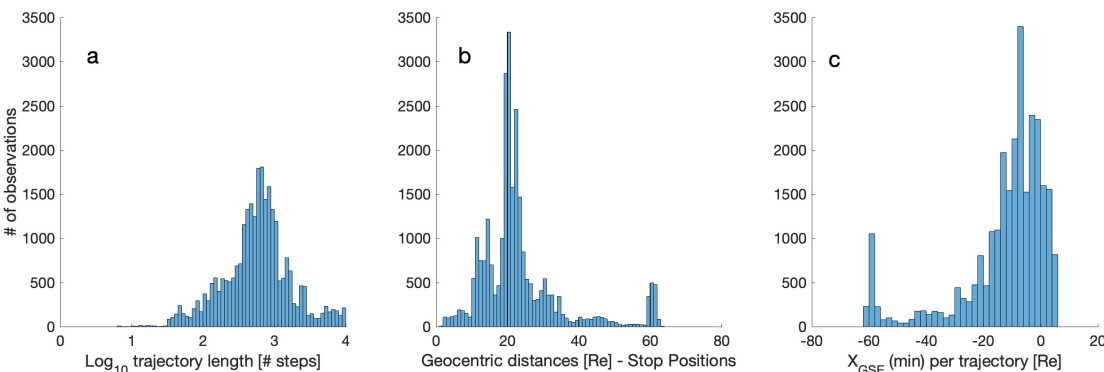

**Figure 4.** (a) Length of the 27200 $O^+$ trajectories in our sample. Note the logarithmic scale. (b) Final positions expressed in the geocentric distance $R$ given in $\mathrm{Re}$ (see text for definition). (c) Minimum X distance for each trajectory.

mantle do not necessarily escape in the distant tail as we suggested in Slapak et al. (2017); Schillings et al. (2019), but they are escaping almost directly through the magnetopause because of their high velocities in these regions. The magnetopause is identified by abrupt changes in the tracing of the magnetic field lines, once the magnetopause is crossed, the field lines become straight and follow the IMF direction. Similarly, we observe 20% of the ions are escaping in the dayside (X > 0 $\mathrm{Re}$). Note that the vertical line of ions at -60 $\mathrm{Re}$ have been stopped tracing due to the limit of our code.

The associated scaled $O^+$ flux (defined as the net outward flux mapped to an ionospheric reference altitude of 1000 $\mathrm{km}$ with a magnetic strength of 50 000 $\mathrm{nT}$) is about $10^{13}$ $\mathrm{m^{-2}s^{-1}}$ in average (not shown). The highest $O^+$ scaled flux, $10^{14}$ $\mathrm{m^{-2}s^{-1}}$, is observed around Earth (-3 Re < X < 3 Re) at $R_{cyl}$ = 23 Re. In contrast, the lower scaled flux is observed below $R_{cyl}$ = 10 Re and between 15 Re < $R_{cyl}$ < 20 Re for X lower than -20 Re.

## 5 Discussion

In our 136 events based on Cluster-CODIF observations, the parallel and perpendicular components of the velocities during the events are taken as inputs to our forward tracing model (see Section 3.2 and Fig. 2). From these observations, we found that $O^+$ ions observed in the plasma mantle or higher altitude cusp have a parallel velocity which is twice the perpendicular component in 93% of the events. More precisely, the ratio between the velocity components ($|v_\parallel|/v_\perp$) is 2.06 $\pm$ 0.83. If we considered that perpendicular velocities measured by CODIF is mainly $\boldsymbol{E} \times \boldsymbol{B}$ drifts, these observations show that $O^+$ ions

at high altitude are not subject to a strong convective electric field anymore. However, Haaland et al. (2007) reported a high plasma convection strongly dependent on the IMF direction and magnitude. In the lobes and for southward IMF, the convection velocities towards the plasma sheet are around 10 $\mathrm{km/s}$ (Haaland et al., 2008). In contrast, the 7% of the our events with higher convection velocity have a corresponding Dst index between -5 $\mathrm{nT}$ and 5 $\mathrm{nT}$ and IMF $B_z$ component between -2 $\mathrm{nT}$ and 2 $\mathrm{nT}$. The highest parallel to perpendicular ratios are found for strong southward IMF (53% of the cases) and strong geomagnetic

disturbances (46% for Dst < -20 $\mathrm{nT}$) (not shown).




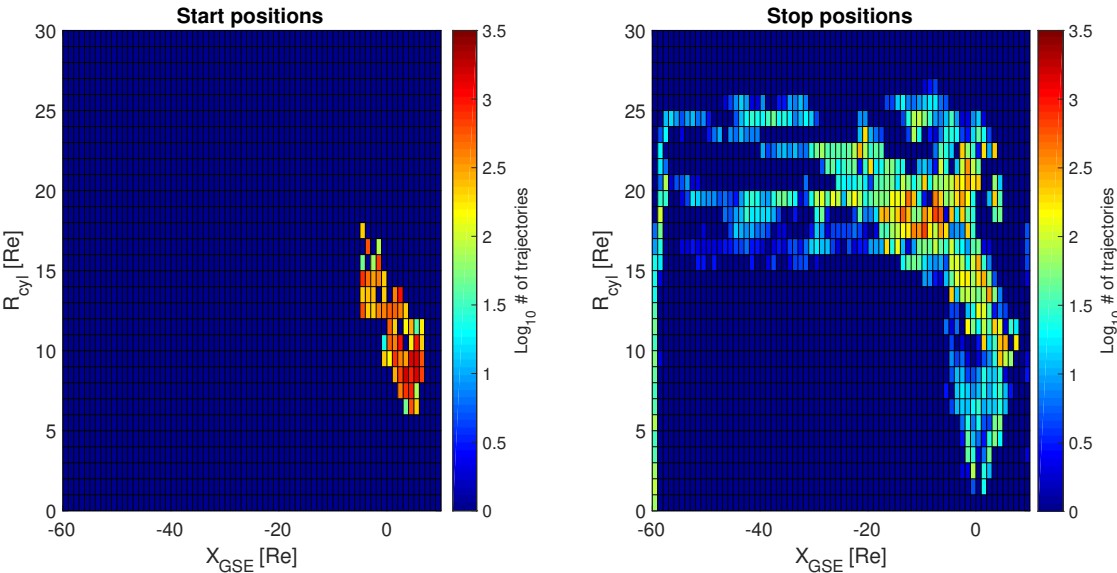

**Figure 5.** Cylindrical coordinates of the starting and ending positions of the launched $O^+$ ions. The colour bar represents the number of trajectories in each bin (1 Re x 1 Re).

We do not find any strong correlation between geomagnetic activity (Dst) and the final positions. For the IMF direction, we identify 47% of the events are associated with northward IMF and the final positions of these ions to be mainly spread between $R_{geoc} = 10$ Re and $R_{geoc} = 35$ Re (82% of the events with northward IMF). A similar trend is observed for the remaining 53% events associated with southward IMF. Thus, the direction of the IMF do not influence in which magnetospheric region the ions

end up. However, if we consider only the ions with their ending positions in $R_{geoc} < 10$ Re, they occur during northward IMF (63%). This result can be compared to the cold ion outflow observed in the lobes during southward IMF. Haaland et al. (2012) found that for southward IMF the cold ion outflow is convected toward the plasma sheet due to strong convection, whereas for IMF directed northward convection is stagnant, so that cold ion outflow reach the far tail.

Slapak et al. (2012) suggested three main routes for ion outflow; (1) cold ion that will end up mainly in the plasma sheet

(Mouikis et al., 2010; Haaland et al., 2012; Liao et al., 2015), (2) energised ions from the cusp to the plasma mantle (Liao et al., 2010; Slapak et al., 2017; Schillings et al., 2019), (3) energised ions from to cusp going directly to the magnetosheath (Slapak et al., 2017). Slapak et al. (2017); Slapak and Nilsson (2018); Schillings et al. (2019) suggested that ions observed in the plasma mantle have sufficient energy and velocity to escape in the distant tail. However, our results show that very few ions reach the distant tail but instead escape directly through the magnetopause after a few minutes ($\sim 22$ min). These $O^+$

ions have short or middle length trajectories in our model (less than 2000 steps, see also Section 4) and represent 89% of our sample. Most (99.3%) of these $O^+$ ions reach a point where the tracing is stopped at a geocentric distance higher than 10 Re and escape the magnetosphere. For ions with trajectories longer than 2000 steps (11% of the total trajectories), 32% is earthward flow due to its interaction with the plasma sheet. Most of these ions do not return to the ionosphere. Some will





instead experience charge exchange, become neutral and be lost from the magnetosphere. This assumption is supported by
Ebihara et al. (2006), who modelled $O^+$ trajectories and introduce a charge exchange process in their model. They estimated
that 2% of the total outflow became neutral due to charge exchange with the hydrogen geocorona. Other particles will drift
to the magnetopause (magnetopause shadowing) and be lost. We note that ion precipitation recorded by the DMSP spacecraft
(Newell et al., 2007) indicates a total precipitation of ions ($H^+$ and $O^+$) of the order $10^{24}$ s$^{-1}$, which is most of the time
dominated by cusp precipitation, not return flow precipitation. This is even less than the return flow estimated by Slapak and
Nilsson (2018), indicating that most return flow indeed does not precipitate to the ionosphere. However, we do not study the
fate of this earthward ions flow and therefore they are not considered as escaping ions in this study.

Under quiet magnetospheric conditions (Dst ≥ -20 nT), it was found that 6% of the final positions of the trajectories is
within a geocentric distance of 10 Re (return flow), whereas during disturbed conditions we observe only 1.5% return flow.
This result agrees with Ebihara et al. (2006), who found that under quiet time 4% to 7% of the outflowing ions return to Earth.
Under disturbed conditions, the authors estimated a smaller return of 0.6% to 0.8%.

Finally, since $O^+$ ions are launched from the plasma mantle, the particles observed by CODIF already went through transverse heating and centrifugal acceleration. Thus this model includes most of the energisation and acceleration compared to
other models. Moreover, the model does not include wave-particles interaction after the oxygen ion has been launched.

## 6   Summary and conclusions

Based on previous suggestions that $O^+$ ions from the plasma mantle are escaping (Slapak et al., 2017; Slapak and Nilsson,
2018; Schillings et al., 2019), we investigate the fate of ions by tracing the particles forward in time in the magnetospshere.
The magnetospshere is represented by the Tsyganenko T96 model for the magnetic field and the Weimer 2001 model for the
electric field (ionospheric potential). We analyse 136 plasma mantle and cusp events detected automatically in the Cluster data
during 2001 and 2007. For each event, 200 $O^+$ ions with an initial parallel and perpendicular velocity are launched from the
plasma mantle or high-latitude cusp. The initial velocities and positions are determined by Cluster observations and are used
as inputs for the forward tracing. Our results are summarised in the following points:

1. The $O^+$ ions observed in the plasma mantle and high-latitude cusp have an initial parallel velocity that is twice the
   perpendicular velocity for 93% of the event. Thus, the parallel velocity dominates from the start, and through high
   perpendicular temperatures, the mirror force will increase the parallel velocity further downstream of the observation
point.

2. The highest ratios between parallel and perpendicular velocities are found for southward IMF (53%) and strong geomagnetic disturbances (46% for Dst < -20 nT).

3. 96% of the final positions (out of 27200) are located beyond a geocentric distance of 10 Re. These particles escape and
   are lost into the solar wind. 20% of the ions escape directly through the high-latitude dayside magnetopause.



4. 3.5% of the total trajectories lead back towards earth, i.e. they constitute return flow. Some of these $O^+$ ions have interacted with the plasma sheet in the distant tail and eventually end up between the Earth and a geocentric distance of 10 Re.

5. Under disturbed magnetospheric conditions (Dst < -20 nT), we observe 1.5 % return flow, whereas during quiet time the return flow increases to 6%.

6. We do not find any correlation between the IMF direction, the geomagnetic disturbances and the final positions of $O^+$ in our tracing model. However, the ions ending up close to the Earth (geocentric distance smaller than 10 Re) are for 63% of the time associated with northward IMF.

*Code and data availability.* The Cluster data can be retrieved from the Cluster Science Archive <https://csa.esac.esa.int/csa-web/>. The solar wind parameters (OMNI data) are provided by the Space Physics Data Facility (SPDF), <https://omniweb.gsfc.nasa.gov>. . Finally,
the tracing code is available for download at <https://doi.org/10.5281/zenodo.3466771>.

*Author contributions.* A.S. made the main analysis and wrote the manuscript. H.N. contributed to the analysis and discussions. H.G and A.D.S. developed the tracing code. R.S., M.Y. and L.G.W. participated in the results discussion. All authors contributed to the writing of the final manuscript.

*Competing interests.* The authors declare that they have no competing interests.

*Acknowledgements.* We thank the CIS team and the Cluster Science Archive team for providing Cluster data. We also thank Tsyganenko, N. and Weimer, D. for sharing their models. Work by HG was supported by the Swedish National Space Agency grant 108/18, by the Belgian Science Policy Office through the Solar-Terrestrial Centre of Excellence, and by PRODEX/Cluster contract 13127/98/NL/VJ(IC)-PEA90316. Finally, we acknowledge the Swedish Institute of Space Physics and the Graduate School of Space Technology hosted by Luleå University of Technology for the financial support.





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
