# Peer review of "The fate of O+ ions observed in the plasma mantle: particle tracing modelling and Cluster observations"

_Annales Geophysicae, 2019_

## Referee Comment (RC1) · Anonymous Referee #1 · 14 Nov 2019

This manuscript presents a test particle tracing study of $O^+$ ions from sites where Cluster observed $O^+$ outflow in the cusp and mantle. The topic is well within the purview of Annales Geophysicae. However, the description of the model omits a number of important details, and there are significant issues with the implementation. Since the implementation issues affect an unknown but possibly substantial fraction of the particle trajectories traced, this is arguably a major revision. Nevertheless, the paper is likely to be publishable once the authors address these issues.

**1. Implementation issues**

[Figure]

The model is based on tracing test particles in a magnetosphere described by the T96 and Weimer 2001 models for magnetic field and potential, respectively. It is implied, but not directly stated, that the authors use the actual solar wind conditions for the events they study. Both T96 and Weimer 2001 are statistical models based on averages of observations under particular solar wind and IMF conditions. Some of the T96 input parameters take into account the history of relevant solar wind and IMF conditions. However, the Weimer models are known to have unphysically rapid responses to shocks and other interplanetary transients, as well as Alfvénic fluctuations in the IMF. Some caveats will be needed in cases where a shock or other interplanetary transient impacts the magnetosphere during the period covered in the modeling.

The model is a test particle model which, as noted in the manuscript, neglects wave-particle interactions. This is a reasonable approximation in the outer magnetosphere but not in the inner magnetosphere, where some combination of chorus, EMIC, and ULF waves can produce pitch angle scattering and therefore affect loss rates (precipitating ions should not be considered losses in the context of this paper). Some caveats on this issue are also needed.

The simulation domain covers $-60 \leq X \leq 10R_E$. The justification for the upper limit is the magnetopause location. But $X = 10R_E$ is merely the average location of the subsolar magnetopause; the actual location can move inward or outward depending on solar wind dynamic pressure. The simulation box should therefore be extended a bit sunward to cover the case of reduced solar wind pressure.

The authors treat the boundary of the inner magnetosphere as spherical and constant in time at a distance of $10R_E$. The real boundary is neither static nor spherical: when solar wind dynamic pressure is elevated the nose of the magnetopause can be pushed inside $10R_E$, which is its nominal location, and the relevant coordinate in the inner magnetosphere is the McIlwain parameter $L$ rather than the radial distance $R$ used in the manuscript.

**2. Clarifications about the model**

Equation (1) as written in the manuscript is not correct, because it has a scalar quantity on the left-hand side and a mixture of scalars and vectors on the right-hand side. If this was intended to be a vector equation, then all terms in the equation should be vectors. If the left-hand side was intended to be a scalar, then the right-hand side should be the square root of the sums of the squares of the components.

Although the reader can deduce this from the subsequent figures, there should be an explicit statement in section 3.1 that the code traces the full 3-D velocity vector rather than using the guiding center approximation.

The authors do not give the $X$ location for the listed magnetopause location of $|Y| = 13R_E$ and $|Z| = 13R_E$. The magnetopause can be approximated as cylindrical in the deep tail but still usually has some flaring at the $X = 0$ plane, which is where I think the quoted numbers are supposed to apply.

In section 3.2, the authors should be explicit about using time-varying solar wind and IMF inputs.

**3. Miscellaneous issues**

The result that the $O^+$ outflow increases by 1.5 orders of magnitude during active times compared to quiet times (specifically using $Kp$ as an activity indicator) is not original with Slapak et al. (2017). The same result was obtained from DE data by Yau et al. (1985), JGR 90, 8417, doi:10.1029/JA090iA09p08417, which paper should be cited in that paragraph.

In the data selection thresholds given at line 119, I think the "and" should be an "or", since the intent is to exclude a velocity range in which the $O^+$ channel is contaminated by protons.

The quantity plotted on the X axis of Figure 4c is described as "minimum X distance"

in the text, the figure caption, and the axis label. I do not think this is an accurate description of the quantity being plotted, since it covers the full range of the simulation box. A clearer description of this quantity would provide better insight into its physical significance.

---

## Referee Comment (RC2) · Anonymous Referee #2 · 23 Dec 2019

This paper reports on a particle trajectory study to determine the fate of particles that have been accelerated in the cusp and located in the outer cusp and plasma mantle. They use observations from 136 events measured by Cluster for the initial conditions, and trace the particles in a Tsyganenko T96 magnetic field and a Weimer 2001 electric field. They find that most of the particles escape out the dusk flank. Only a small minority come to the inner magnetosphere. Overall the work is well done. Below I list some specific questions that should be addressed. Although most of these could be considered minor, they could also end up having major consequences, so I have checked "major" above since "minor" implied the paper would not go out for re-review.

[Figure]

1) Line 170 - I don't understand the statement, "the perpendicular component of the velocity corresponds to the general variability of the data set". Are you saying that it is mainly statistical error represented here? It does seem that the thermal velocity would be more appropriate to use than the bulk perpendicular velocity, since it is mainly the pitch angle of the particle that is important for the trajectory, since the convection used is from the Weimer field. A distribution with a large pitch angle, could have net zero perpendicular velocity, but it would certainly lead to a higher parallel velocity along the trajectory, as you state in the next line. Can you explain better why you made the choice you did?

2) How good these trajectories are clearly depends on how good the field models used are. I suspect that the trajectories are quite sensitive to the initial conditions in the "start" location. The paper should include some discussion about how well the T96 model does in this region, and how sensitive the trajectories might be to discrepancies. The paper, Tsyganenko and Russell, 1999 implies that there needs to be additional corrections to correctly model the cusp. Has anything like this been included? Have the fields in the observation location been checked against the Tsyganenko fields, to make sure that they are in reasonable agreement? Has it been checked that the initial positions are in the "cusp" region of the Tsyganenko field?

3) For the example shown in Figure 3, 196 out of 200 are "long trajectories". However, in general, ion with "long trajectories" are only 11% of the sample, implying that the sample shown is not representative. Is this because of the chosen initial position? How much does the type of trajectory depend on the initial location? Perhaps a figure like Figure 5a (maybe blown up around the region of interest) that shows the average trajectory length in each bin would help clarify this? Or a set of 3 figures that show number of long trajectories, number of medium, and number of short in each bin? Or perhaps better would be figures that show the eventual fate of ions from different start locations (down tail, out dusk, into inner magnetosphere).

Line 209-215 - It is not clear what is meant by the statement "We define the escaping

limit by the distance of the final position R=10 Re." I suggest rewording to "An ion is defined to have "escaped the magnetosphere" if its final position is outside R=10."

Line 216 - Although this line is technically correct, I suggest rewording to make it clearer. "We determined the MAXIMUM distance in |X| for each trajectory".

Lines 281 and 282, Magnetosphere misspelled twice.

Reference:

Tsyganenko, N. A., & Russell, C. T. (1999). Magnetic signatures of the distant polar cusps: Observations by polar and quantitative modeling. Journal of Geophysical Research, 104(A), 24939–24956. http://doi.org/10.1029/1999JA900279

---

## Author Comment (AC1) · 25 Feb 2020

**Answer to the referee #1**

We would like to thank the referee for his/her comments. We hereby answer the comments in green.

By going through the revisions, we found some errors in the automatic routine as well as the value of the perpendicular velocity in equation 1. The perpendicular velocity used in the submitted manuscript has been the bulk perpendicular velocity calculated from CODIF instrument. However, this perpendicular velocity is dominated by the ExB drift velocity, to what we added an additional initial ExB at the start of the tracing. Even though this was done, the velocity was underestimated compared to the thermal velocity of ions (for CODIF, the estimated thermal velocity is significantly higher than the bulk perpendicular velocity). We should have used the thermal velocity to begin with. We therefore reran all our events with the thermal velocity instead of the perpendicular velocity. We consequently updated all figures accordingly to the new simulations. The results are very similar but slightly different. Please also note that we removed "cusp" from the title to avoid confusion, since our observations are mainly in the plasma mantle region.

1. Implementation issues

The model is based on tracing test particles in a magnetosphere described by the T96 and Weimer 2001 models for magnetic field and potential, respectively. It is implied, but not directly stated, that the authors use the actual solar wind conditions for the events they study. Both T96 and Weimer 2001 are statistical models based on averages of observations under particular solar wind and IMF conditions. Some of the T96 input parameters take into account the history of relevant solar wind and IMF conditions. However, the Weimer models are known to have unphysically rapid responses to shocks and other interplanetary transients, as well as Alfvénic fluctuations in the IMF. Some caveats will be needed in cases where a shock or other interplanetary transient impacts the magnetosphere during the period covered in the modelling.

We use indeed the actual solar wind for the events studied. Most of strong disturbed conditions are removed from the data by Tsyganenko model requirements. Additionally, we looked at shocks through the Cfa interplanetary shocks database (https://www.cfa.harvard.edu/shocks/) simultaneously with the solar wind data. After a cross-check of these data, we removed 5 events. If a shock appeared in the next hours (after our events), we did not remove the event considering that if the solar wind conditions were too extreme at the initial time of our event, Tsyganenko requirements does not take the event into account. See clarifications in lines 132-133 of the revised manuscript.

The model is a test particle model which, as noted in the manuscript, neglects wave-particle interactions. This is a reasonable approximation in the outer magnetosphere but not in the inner magnetosphere, where some combination of chorus, EMIC, and ULF waves can produce pitch angle scattering and therefore affect loss rates (precipitating ions should not be considered losses in the context of this paper). Some caveats on this issue are also needed.

Our simulations are done mainly in the plasma mantle, which is considered to be in the outer magnetosphere (and very few events from the high-latitude cusp). In our automatized routine, we used a criteria of R > 6 Re in order to remove the inner magnetosphere. Therefore, we believe that our code is valid and gives reasonable trajectories for the ions observed in the

plasma mantle. Also, our main purpose is to confirm that majority of the O⁺ in the plasma mantle are directly escaping into the solar wind or in the distant tail. Should our tracing lead the particles to the inner magnetosphere, we consider them as not lost, as discussed in the paper.

The simulation domain covers 60 <= X <= 10 Re . The justification for the upper limit is the magnetopause location. But X = 10 Re is merely the average location of the subsolar magnetopause; the actual location can move inward or outward depending on solar wind dynamic pressure. The simulation box should therefore be extended a bit sunward to cover the case of reduced solar wind pressure.

The simulation box is used for the electric field calculation along the magnetic field lines. Since we are tracing the ions from the plasma mantle and very few in the cusp, the limit of the electric field box is enough at X = 10 Re. The Y and Z directions are more important from X = 0 Re to approximately X = 5 Re, in order to include the moving magnetopause in the Y and Z directions. This aspect is covered by our limits in |Y| and |Z| directions that equals approximately 20 Re. The ions traced in the dayside plasma mantle are escaping almost directly through the magnetopause and never reach distances higher than the simulation box limit in the dayside. Therefore, the limit at 10 Re in X direction is enough for our study and extending the box would require more computational time without providing more accurate trajectories.

The authors treat the boundary of the inner magnetosphere as spherical and constant in time at a distance of 10Re. The real boundary is neither static nor spherical: when solar wind dynamic pressure is elevated the nose of the magnetopause can be pushed inside 10 Re, which is its nominal location, and the relevant coordinate in the inner magnetosphere is the McIlwain parameter L rather than the radial distance R used in the manuscript.

We agree that the representation we used do not represent the real boundary. However, we only want a parameter that defines the boundary where the ions escape. In our case, we believe a spherical and static boundary for average conditions is good enough for statistics. For slightly disturbed conditions, the whole magnetosphere is compressed and our boundary is then overestimated and ions escape anyway. Please also note that strong disturbed conditions are removed by Tsyganenko model requirements.

2. Clarifications about the model

Equation (1) as written in the manuscript is not correct, because it has a scalar quantity on the left-hand side and a mixture of scalars and vectors on the right-hand side. If this was intended to be a vector equation, then all terms in the equation should be vectors. If the left-hand side was intended to be a scalar, then the right-hand side should be the square root of the sums of the squares of the components.

We are sorry for the confusion, equation (1) is a vector. We have corrected the equation in the new manuscript. Please note that we have updated the equation according to our new simulations (thermal velocity instead of bulk perpendicular velocity), lines 150-154. See also explanation at the beginning of the review.

Although the reader can deduce this from the subsequent figures, there should be an explicit statement in section 3.1 that the code traces the full 3-D velocity vector rather than using the guiding centre approximation.

We slightly changed the text, see lines 150-154. Additionally, equation (1) has been rewritten and therefore clarify the 3 dimensions of the velocity vector.

The authors do not give the X location for the listed magnetopause location of |Y | = 13 Re and |Z| = 13R E. The magnetopause can be approximated as cylindrical in the deep tail but still usually has some flaring at the X = 0 plane, which is where I think the quoted numbers are supposed to apply.

Yes, the numbers for Y and Z apply for X=0. We have added this detail in the new manuscript, see line 162.

In section 3.2, the authors should be explicit about using time-varying solar wind and IMF inputs.

We added a precision in the sentence saying that the solar wind conditions for each corresponding event are taken at the initial time (start time of the event). See lines 171-172 in the reviewed manuscript.

3. Miscellaneous issues

The result that the $O^+$ outflow increases by 1.5 orders of magnitude during active times compared to quiet times (specifically using Kp as an activity indicator) is not original with Slapak et al. (2017). The same result was obtained from DE data by Yau et al. (1985), JGR 90, 8417, doi:10.1029/JA090iA09p08417, which paper should be cited in that paragraph.

We have added this reference and Yau et al. (1988) as well, see lines 43-46. Slapak et al. study is actually based on their flux equations. The main difference between Slapak et al (2017) and Yau et al. (1985, 1988) is altitude. Slapak et al. examined the O+ ions in the plasma mantle whereas Yau et al. examined lower altitudes (accordingly, the energy range is different) and this is why our simulations start from the plasma mantle rather than DE altitude that all past models used.

In the data selection thresholds given at line 119, I think the "and" should be an "or", since the intent is to exclude a velocity range in which the O + channel is contaminated by protons.

Yes, indeed the "and" should be changed in "or". We have corrected this in the new manuscript.

The quantity plotted on the X axis of Figure 4c is described as "minimum X distance" in the text, the figure caption, and the axis label. I do not think this is an accurate description of the quantity being plotted, since it covers the full range of the simulation box. A clearer description of this quantity would provide better insight into its physical significance.

We defined the minimum distance Xmin as the minimum value in the X direction of the trajectory length. So in a trajectory of 280 steps, we take the minimum value in X direction within the 280 points. The maximum number of steps for a trajectory is 10000 and the shortest trajectory we obtained has 7 steps, the average trajectory steps is 1029. So, figure 4c represents the minimum value of each trajectory length, which indicates that most of the ions reach distances between X = -10 Re and X = -20 Re. These ions might end their journey at that distance or may go back towards Earth after interacting with the plasma sheet. For most of the particles the minimum distance is their ending position. The peak at X = -60 Re includes the particles stopped at the limits of our model.

We have now clarify this "minimum X distance" in the new manuscript, see lines 224-227.

---

## Author Comment (AC2) · 25 Feb 2020

**Answer to the referee #2**

We would like to thank the referee for his/her comments. We hereby answer the comments in green.

1) Line 170 - I don't understand the statement, "the perpendicular component of the velocity corresponds to the general variability of the data set". Are you saying that it is mainly statistical error represented here? It does seem that the thermal velocity would be more appropriate to use than the bulk perpendicular velocity, since it is mainly the pitch angle of the particle that is important for the trajectory, since the convection used is from the Weimer field. A distribution with a large pitch angle, could have net zero perpendicular velocity, but it would certainly lead to a higher parallel velocity along the trajectory, as you state in the next line. Can you explain better why you made the choice you did?

We originally used the bulk perpendicular velocity calculated with CODIF instrument, which is dominated by the ExB drift, which is included in the model and should not have been added to the particles. Just as the reviewer notes, we should have used the thermal velocity. We re-did the whole analysis with the thermal velocity instead. Since the thermal velocities where higher than our initial perpendicular velocity (1st manuscript), the total velocity was slightly underestimated. However, the final result did not change significantly.

See the general comment at the top of the answers and also equation (1) and paragraph 3.2 in the reviewed manuscript.

2) How good these trajectories are clearly depends on how good the field models used are. I suspect that the trajectories are quite sensitive to the initial conditions in the "start" location. The paper should include some discussion about how well the T96 model does in this region, and how sensitive the trajectories might be to discrepancies. The paper, Tsyganenko and Russell, 1999 implies that there needs to be additional corrections to correctly model the cusp. Has anything like this been included? Have the fields in the observation location been checked against the Tsyganenko fields, to make sure that they are in reasonable agreement? Has it been checked that the initial positions are in the "cusp" region of the Tsyganenko field?

The trajectories are indeed sensitive to the initial conditions. However, we checked the magnetic field provided by Cluster at the starting positions and the corresponding magnetic field calculated by Tsyganenko model. We found that both magnetic field correspond pretty well to each other even though an increase in magnitude is observed closer to the Earth, see Fig. below (not shown in the manuscript). The blue and red components correspond to Cluster data and Tsyganenko respectively. We also checked the tracing of the lines individually for each event and we found that 4 events out of the 131 were most probably in the cusp (3%). Consequently, to avoid confusion for the reader, we removed "cusp" from the title and we focused our manuscript on the plasma mantle region.

3) For the example shown in Figure 3, 196 out of 200 are "long trajectories". However, in general, ion with "long trajectories" are only 11% of the sample, implying that the sample shown is not representative. Is this because of the chosen initial position? How much does the type of trajectory depend on the initial location? Perhaps a figure like Figure 5a (maybe blown up around the region of interest) that shows the average trajectory length in each bin would help clarify this? Or a set of 3 figures that show number of long trajectories, number of medium, and number of short in each bin? Or perhaps better would be figures that show the eventual fate of ions from different start locations (down tail, out dusk, into inner magnetosphere).

With our new events (131 instead of 136) the long trajectories are reduced to 5% of the sample. The starting points of the long trajectories do not influence how long the trajectory will be. We choose this particular example for the paper to illustrate the 3 types of trajectories even though long trajectories represent only 5% of the sample. To better illustrate the different trajectories we replace Figure 5 by a similar figure but divided in short, middle and long trajectories instead. We discuss this new figure lines 233 – 237 in the revised manuscript.

Line 209-215 - It is not clear what is meant by the statement "We define the escaping limit by the distance of the final position R=10 Re." I suggest rewording to "An ion is defined to have "escaped the magnetosphere" if its final position is outside R=10."

We have added the sentence to clarify our definition of the escaping boundary, see lines 218-219.

Line 216 - Although this line is technically correct, I suggest rewording to make it clearer. "We determined the MAXIMUM distance in |X| for each trajectory".

Since this parameter was not clear, we rewrote the sentences, see lines 224-227. See also comment #1.4 and #3.3 of referee #1. We defined the minimum X distance as the smallest value in the X direction for each ion trajectory. This parameter help to understand if ions move directly to the magnetopause or experience return flow (and interact with the plasma sheet) starting from the plasma mantle region. Therefore it is not the maximum distance |X|.

Lines 281 and 282, Magnetosphere misspelled twice.

Thank you, this has been corrected.

---

## Author Response (AR2)

First, we would like to thank the referees for their comments that help improving the final manuscript. We wrote the answers to the comments/suggestions in green in the following document.

**Referee #1**

Overall, the author has addressed my concerns.

However, a better display of the initial and final positions of the trajectories would make the results clearer. Figure 3 gives a much more understandable picture of what is going on than Figure 5. And it is clear from Figure 3 that Y and Z are not equivalent, and positive and negative Y are not equivalent. So rolling all those things up into R-cylindrical hides most of the interesting information. Instead of this display, could the start and end positions be displayed in Z vs X and Y vs X for the three trajectory ranges, similar to Figure 3? I don't think Z vs Y is needed, but the authors could check if it shows anything interesting.

Adding a figure, or adding a few additional plots to Figure 3 with sample short trajectories and middle trajectories would also help.

I realize the main point of the article is just to summarize how many ions are lost, but having run all of these trajectories, it would be better to show where exactly the particles end up in a more illuminating way.

We added a plot (Figure 4 in the revised manuscript) with a sample of short, middle and long trajectories as suggested. We updated the main text accordingly; see lines 211-213 in the revised manuscript.

In addition, I have the following minor comments:

Lines 21-22. I don't understand the relationship between the first clause of this sentence and the second clause. What does the term "ionospheric outflow" have to do with distinguishing solar wind H+ from ionospheric H+?

The polar wind includes electrons, H+, O+ and He+. At higher altitude, the compositions change significantly and in the F region O+ ions dominate. Therefore, at Cluster altitude (several Re), the observed O+ has an ionospheric origin. At the same altitude, the observed H+ could be either from ionospheric origin or from solar origin. Therefore, H+ outflow (with a terrestrial origin) is more difficult to estimate than O+. We modified the sentence to make it clearer, see lines 21-22.

Line 42 - I think "polar cap" in this sentence should be "polar cap and auroral zone". The Yau results really focused on the auroral zone (which includes the cusp).

We added "the auroral zone" in the sentence, please see line 43.

Line 266 - Should be "the cusp" not "to cusp"

We have now corrected the mistake, thank you.

**Referee #2**

1.The middle panel of Figure 4 does not contribute to the paper. The description of this panel is inadequate, and the points that the authors intend to make with this panel are made much more clearly with Figure 5.

We do believe that the middle panel contributes to the paper because it shows our definition of the escaping ions. All the ions ending at a geocentric distance higher than 10 Re are eventually escaping, thus this panel illustrates our main result (98% of the ions from the plasma mantle are escaping). To make it clearer, we slightly modified the panel by adding the limit of escaping ions and a short explanation textbox.

2. The claims made in the second paragraph of the Discussion section (beginning on line 254) need to be supported with figures. This is the only place in the paper where any mention is made of the correlation of final positions, or lack thereof, with geomagnetic and interplanetary conditions. The principle of "show, don't tell" applies here.

We added the figure, which the text was based on. Please see Figure 7 in the revised manuscript. We also updated the main text accordingly, see lines 255-260 and 282 in the revised manuscript.

3. A technical correction is needed in the fourth sentence of the above mentioned paragraph. A better version would be: 
[revised manuscript text omitted]